# User concerns regarding information sharing on social networking sites: The user's perspective in the context of national culture

Ibrahim Mutambik[1]ʘ, John Lee[2]ʘ*, Abdullah Almuqrin[1]ʘ, Waleed Halboob[3]‡, Taha Omar[4]‡, Ahmad Floos[5]‡

1 Department of Information Science, King Saud University, Riyadh, Saudi Arabia, 2 School of Informatics, the University of Edinburgh, Edinburgh, United Kingdom, 3 Centre of Excellence in Information Assurance (CoEIA), King Saud University, Riyadh, Saudi Arabia, 4 Deanship of Development and Quality, King Saud University, Riyadh, Saudi Arabia, 5 Deanship of Skills and Development, King Saud University, Riyadh, Saudi Arabia

ʘ These authors contributed equally to this work.
‡ WH, TO and AF also contributed equally to this work.
* john.lee@ed.ac.uk

**Data Availability Statement:** All relevant data are within the paper and its Supporting Information files.

## Abstract

Information sharing via social networking systems (SNS) is a common practice among academics, as well as others, that brings substantial benefits. At the same time, privacy concerns are widespread among SNS users, which may tend to inhibit their maximising the benefit from using the systems. This paper investigates the proposition that SNS user attitudes and behaviour are affected by privacy concerns, and that the effects are subject to significant cultural factors. A broad assessment of the literature provides the context for the study. Working in the context of Saudi Arabia, we apply a mixed-methods approach beginning with in-depth interviews, exposing in detail a range of views and concerns about privacy and SNS use, also allowing us to identify three key factors that bear on SNS usage and users' concerns. Analysis of these factors in the light of the "theory of reasoned action" derives a structural model predicting several hypotheses relating the factors and users' attitudes and behaviour. We assess the model through development of a questionnaire, administered to a large pool of academic participants, that allows us to examine how the model responds in general, and via multigroup partial least squares analyses, differentially to gender and to culturally distinct (Arab vs. non-Arab) constituents of the participant group. Results show good support for the hypotheses and clear gender and culture effects. Picking up issues from the interviews, discussion focuses on users' views about SNS providers' privacy policies and their inadequacy regarding culturally specific ethical concerns. We argue that these views may reflect different regulatory environments in combination with other cultural factors.

**Funding:** The authors extend their appreciation to Researchers Supporting Project number (RSP-2021/233), King Saud University, Riyadh, Saudi Arabia.

**Competing interests:** The authors have declared that no competing interests exist.

# Introduction

Practices of sharing information via social networking services are pervasive and very diverse. Many different services exist, by which huge amounts of information are shared for purposes ranging from entertainment to professional development. We are especially concerned, in this paper, with information sharing by academic users of social networking services (SNS), and there are very diverse practices even here, differences that may be individual or related to particular disciplines, etc. Information may be shared between academics, or between academics and the wider public. There are many benefits, but also many concerns. For example, the reliability of information that is found via SNS is often in doubt: in the "post-truth" world, there is a constant battle against misinformation, especially in areas such as medicine [1]. Other concerns that arise around information sharing, and SNS use in general, involve issues of privacy. Information may be appropriated, misused, stolen; exploited without permission for the gain of others. Among the many confusing and perhaps paradoxical aspects of this situation, key questions continue to surround the nature of user behaviour and how it is influenced [2–4]. Attitudes and behaviours in relation to information privacy form the focus of this paper.

It is often held that users do not behave in ways that would optimise the safe use of information sharing systems. They tend either to act conservatively, failing to realise the potential benefit of the systems [5], or else to act rashly and risk having their information stolen and misused [6, 7]. In general, little is known about the factors driving behaviour and how these might be addressed. However, there is emerging evidence that cultural factors may play an important role [5, 8, 9].

In this paper, we investigate behavioural factors in relation to certain cases of information sharing in Saudi Arabia. In doing this, we seek to identify the purposes and motivations of information sharing, the factors that condition behaviour, and we consider how a model analysing these factors needs to be interpreted in the light of their interactions with the cultural context.

Behaviour arises as an outcome of complex mechanisms that can be characterised in many different ways. In this paper, we adopt the lens of the Theory of Reasoned action [10] which focuses especially on the role of attitudes and intentions in conditioning behaviour. We therefore seek to identify factors that can be recognised in terms of their effect on attitudes and intentions. A deeper understanding of how individuals perceive privacy issues will lead to valuable insights for practitioners and academics alike. Although there are numerous pioneering studies that examine online privacy in general [11–14], there is a lack of studies that have systematically attempted to provide a theoretical framework on the specific nature of information privacy, particularly from the perspective of SNS users. This paper seeks to fill that gap by exploring the following questions:

1. What are the critical information privacy factors that have a significant influence on SNS users for information sharing?

2. To what degree do the information privacy factors influence SNS users' information sharing behaviour?

3. How are the answers to these questions subject to cultural context?

Altogether, this paper provides a deeper understanding of users' information privacy concerns on SNS for information sharing. The paper is structured as follows. Section 2 provides a review of the background in terms of information privacy concerns against which we construct our study. Following this, the research methodology, analysis of the data, and findings are presented. Lastly, the practical implications and contributions to theory are given.

## Background: Knowledge sharing, information and privacy

### Knowledge sharing

Defining knowledge sharing is important, particularly with the growing use of social networks as a means of knowledge sharing. According to Wang and Noe [15, p.117], the term knowledge sharing is defined as "the provision of task information and know-how to help others and to collaborate with others to solve problems, develop new ideas, or implement policies and procedures". Chen et al. [16] also defined knowledge sharing as a process of exchange between people, groups, or organisations, while Pangil and Chan [17, p.52] define it as "the communication of knowledge from a source in such a way that it is learned and applied by the recipient". Knowledge sharing takes place when an individual conveys their knowledge to others, or obtains it from them [16]. Social networks enable the transmission of knowledge between individuals from all parts of the world at a much faster rate than more traditional methods, for example databases or search engines [17].

Nowadays, many academics, students and businesses have focused their attention towards social networks due to the ability of social networks to enable users to share knowledge across a range of different knowledge fields, such as shopping, culture, arts, sports, and business, among others [18, 19]. There are many roles of social networks, including allowing users to rapidly and easily integrate their own knowledge and ideas with those of others, anywhere and at any time [20]. Social network platforms also present a hugely effective virtual space in which knowledge can be shared and acquired by many individuals, groups, and communities [21]. Further, these platforms are acknowledged as key tools that enable the connection of many users in one place, who can quickly and easily share their thoughts, experiences, and knowledge with one another [20]. In many areas, such as the economy and medicine, SNS have rapidly become one of the key tools whereby people find information, for example on diseases and treatments in neurology [22]. Again, this emphasises the importance of accuracy and helping people to recognise information that is reliable.

Research by Al Saifi et al. [18] found that social network use helps users to more easily find useful information within their organisation. This is supported by Al-Qadhi et al. [8], who show that employees are often encouraged by their employers to use social networks for knowledge sharing as it is the most efficient way to disseminate knowledge and information. IBM also utilises a social network for sharing knowledge among its 400,000 workers, which facilitates collaboration within the company and keeps them innovative [19]. Moreover, research has revealed that 86 percent of managers perceive social networks to be a beneficial way to share knowledge in their companies [23].

The largest social network providers give users a platform from which they can connect and interact with many hundreds, if not thousands, of other users around the world, exchanging information and generating online content [21]. Social networks are now the main communication channel for knowledge and information sharing, with users connecting with family, friends, businesses and workplaces [4]. More broadly, SNS can provide a social support mechanism, being perceived by users to facilitate health-related coping and social interaction, and may suggest a new kind of therapeutic alliance between physicians and people with chronic diseases in medicine [24].

A review of the relevant literature revealed three central activities that take place on social network platforms, which make them distinct from older types of communication and knowledge sharing behaviours: namely social interaction, seeking knowledge, and contributing to knowledge.

Studies have shown that when staff use a social network to share knowledge within their organisation, it is crucial that they utilise it correctly in order to maximise the benefit and help

generate new knowledge [25]. Yu and Yu [26] explain that the more individuals use social network for knowledge sharing within their workplace, the greater their contributions are to the organisation as they can more effectively exchange valuable information and knowledge among their peers. Zeng and Zhong [27] describe knowledge contribution as the degree to which an individual's imparted knowledge is useful to others. Knowledge contribution that takes place on social network platforms is observable in many forms, such as a user posting important or useful information, or responding to questions asked by other users with answers, opinions, feedback or solutions, all of which are positive, knowledge-based interactions [18]. There even exist specific social network applications with the sole purpose of allowing users who require knowledge for work to easily find and share what they need [19].

The studies mentioned thus far have covered areas including the relationship between information privacy and the field of internet usage in general. However, to date there seems no attempt to explore the impact of national culture on the relationship between information privacy and using social networks as a means of knowledge sharing.

Culture as a main influencing factor in people's information privacy concerns was first determined by Milberg et al. [28], who also found that the way a country regulates the management of information security and privacy depends on the extent to which its citizens are concerned with the issue. Moreover, an exploration of how culture has an impact was carried out by Bellman et al. [29] across 38 sample countries, and the findings showed that the culture of a society significantly impacted individuals' perceptions of information privacy. Likewise, a comparative study of ecommerce technology adoption between the United States (US) and Italy showed that US society, which is individualistic in nature, tends to have a higher degree of trust compared to the more collectivistic, low-trusting Italian society. This was evidenced by the difference in calculus that consumers from each nation go through prior to making online transactions.

## Privacy

Concerns regarding information privacy include people's perceptions and attitudes towards how their information is treated and shared with others [30, 31]. A person's concerns over privacy are influenced to a significant degree by external factors such as the industry sector, culture, and a country's rules and regulations [32]. In addition to this, the perceptions of these external influences can also vary depending on a person's characteristics and prior experiences [31–34]. Thus, individuals often have very different opinions and attitudes towards the fairness (or lack thereof) of the collection of their personal data and the way it is used and shared.

Researchers use a range of scales to measure individuals' information privacy concerns, and such measures are commonly a one-dimensional global privacy concern (GPIC) type [35–37]. While GPIC does assess privacy concerns quite generally, it is not able to evaluate the particular dimensions of these concerns in any great depth. For this reason, Smith et al. [36] conducted a range of studies utilising rigorous methodologies to reveal more details regarding information privacy concerns. The result of this was the creation of a novel multidimensional scale, which the authors named 'concern for information privacy' (CFIP), and this scale can be used to more accurately measure privacy concerns of individuals in the context of data used by organisations. The CFIP consists of 15 items that reflect four key dimensions of privacy concern: improper access; collection; unauthorised secondary use; and errors. Stewart and Segars [38] then identified the CFIP scale's psychometric properties and empirically confirmed them.

Ever since the establishment of the CFIP model and its acceptance as a reliable and valid measurement instrument, many research projects have successfully applied it in fields such as internet and application usage, offline direct marketing, and social media marketing [34, 35, 37, 38]. Nevertheless, "the dimensionality is neither absolute nor static, since perceptions of

advocates, consumers, and scholars could shift over time" [36, p.190]. This has been clearly evidenced in the widespread use of the internet around the globe and, subsequently, in the significant changes to the ways in which businesses are now run. An example of this is how consumers are able to control their personal information that is stored about them in a company's database, which is not the case for traditional media.

An alternative approach to the psychometric model is to focus more on behaviour. However, people's behaviour can be quite complex and, for example, vary between different social media contexts. Aghasian et al. [39] offer a scoring system that derives a "privacy disclosure score" for users, taking into account that they may use several social networks in different ways. Papaioannou et al. [40] identify a distinction between "social threats" and "organisational threats", where only concern about the former has an impact on disclosure of personal information; they find also that a user's behaviour differs between different social media platforms, and that there is a connection between self-esteem and the amount of personal information revealed. Concern about privacy thus emerges as a rather variable element in a mesh of factors affecting behaviour.

As noted in a recent review by Mitchell and El-Gayar [7], a focus of previous literature has often been the extent of users' knowledge of the "privacy policies" of service providers. It is sometimes claimed that users who know these better are more likely to share information, and sometimes that they are likely to be more wary. On the other hand, it is also claimed that users essentially never read the Terms of Service of service providers, and generally have no direct knowledge of their privacy policies [41]. This neglect of policies is widely deplored and often held up as a paradigmatic example of the "privacy paradox", whereby users who claim to be concerned about privacy nonetheless show scant regard for it in practice. One of our interests is to investigate how users view the importance of privacy policies and whether or how national culture bears on this.

The users in our study are all academics, using social networks for sharing knowledge. Therefore, the question arises whether these are different from users in other social networking contexts. In recent years, a burgeoning literature on "academic social networking" has focussed largely on sector-specific services such as ResearchGate, Academia.edu etc., rather than examining academic uses of more general social networks such as Facebook and Twitter. This literature has rarely had much to say about privacy concerns, except to note that they exist and may be a hindrance to wider adoption of the services [42–46]. Asmi and Margam [46] found that, although data security was the "main concern" of academic network users, in common with many other studies, the greatest hindrance to using them more is simply lack of time.

Although the users in our study are generally not using these specifically academic networks, there appears to be much commonality between users across types of networks in terms of concerns and behaviour regarding privacy. Salahshour et al. [42], for example, reveal that academic network users rarely read the privacy policies of their networking sites, with statistics broadly comparable to those that Obar and Oeldorf-Hirsch [41] find for more general users.

While it could be that academics form a distinctive subculture of social networking users, there is a question about how this combines with national culture. Perhaps, for our participants, the cultural context of Saudi Arabia is even more significant than the academic context. An investigation by Alsolamy [47] is especially relevant, being congruent with ours in focussing on Saudi academics who use general social networks such as Facebook and Twitter for knowledge sharing. Alsolamy [47] finds that there is a very high concern with privacy issues among these academics, especially women. In itself, this is not unlike findings in other studies mentioned above, but Alsolamy [47] argues that in Saudi culture there is a very deep cultural

value attached to privacy, and to e.g. the control of images of an individual, especially among women, going as far as to suggest that "privacy" has a different meaning in different cultures [47]. Cannataci [48, p.5] traces Islamic concepts of privacy to roots in scripture, while also noting: "many scholars within different jurisdictions have engaged in a continuing debate about the definition of privacy but while this debate has yielded many valuable insights we are nowhere closer to having a universally agreed definition for a universal value such as privacy". Hence, privacy may be a concern where differences in cultural sensitivity show up more than some other differences between user populations.

There is a notable strand in the literature concerning the relevance of cultural context to behaviour in regard to privacy. There are strong arguments that attitudes and practices towards privacy and private information are dependent on cultural factors. It seems to be widely established that there are cultural effects on attitudes and behaviour in relation to privacy, for instance between individualistic and collectivistic cultures. Usually these are associated with different countries, as exemplified in the typical application of the Hofstede dimensions [49]. Occasionally, it is noted that there may be cultural differences between regions within a country [18]. However, there may also be cultural differences within a population even where there is no geographical separation. We investigate this proposition with relation to the cultural differences between the Arab and non-Arab sectors of the population in Saudi Arabia.

Abokhodair et al. [50] develop the insights that there are quite specific understandings of privacy in Muslim societies, and also that cultural expectations have a profound effect on social media use, in particular the need to control access to information (especially for women). This is seen as arising partly from the relatively collectivist nature of these societies, and the tendency for social norms to be enforced through deeply shared practices and expectations. A possible corollary would be that those who do not share, at least to the same degree, this culture, would be differentiated in terms of their attitudes to privacy. We therefore conjecture that the non-Arab elements of the Saudi population will show differences in these respects.

## Research method

Our study, as mentioned, is situated in Saudi Arabia, which presents a relatively specific cultural context, and within that context we are working with users with particular information sharing concerns. The empirical element of this study, and aspects of the data collected, has already formed the basis of a separate discussion specifically about the gender differences that can be found in users' concerns and behaviours [51]. In this paper, we discuss separate aspects of the same data; there is therefore some overlap in the treatments of the setting and methodology, but not the analysis, discussion or conclusions. In common with that account, we proceed first to investigate the overall concerns of this user group, identifying factors that we can then study in more depth. This focus allows us to be more precise in our observations; we later discuss the considerations that arise in drawing more general conclusions from the study.

This study utilises a mixed methods approach to research, which according to Johnson and Onwuegbuzie [52, p. 17] is where "the researcher mixes or combines quantitative and qualitative research techniques, methods, approaches, concepts or language into a single study". A mixed methods approach was chosen because the use of both quantitative and qualitative methods–drawing on the strengths of each–can provide rich and comprehensive findings that can be missed if only one approach is focused upon. This was carried out in two separate stages so that the research questions could be addressed and answered.

The first stage was exploratory, using a qualitative approach to examine and obtain a richer understanding of the critical information privacy factors that have a significant influence on SNS use for information sharing in our particular user group, as well as establishing the study's

research model and hypotheses. From the viewpoint of the user, these hypotheses were unknown at the start of the research. Every step of this exploratory stage used findings from existing literature for provision and to support with rational reasoning–particularly when identifying behavioural information privacy factors that influence SNS users for information sharing–as well as for hypothesis development. Further details of both stages are given below and in Mutambik et al. [51].

Once the first stage was complete, the study moved on to the second stage: development of a confirmatory factor analysis. A quantitative questionnaire was developed for use in this stage, based on the outcome of the earlier research stage. The research model was tested and validated using the questionnaire, along with any factors related to information privacy that may have an influence on SNS users. At this stage, we were also able to identify a comparison between two cultural groups: the Saudi Arab majority, on the one hand, and a more diverse immigrant population on the other. Mutambik et al. [51] note (in Table 2, p.245) that both "Native" and "Non-native" participants were included, but do not otherwise mention this distinction, whereas here it forms the major focus.

Before data collection, the study was reviewed and approved by the Research Ethics Committee at King Saud University. Before starting the data collection, all participants were asked to read an informed consent form, which explained that any information they provided would be anonymized and kept in confidence, that data would be used solely for purposes of the study, and that all information would be destroyed when the study was completed. They were informed that they could withdraw consent to participate at any time without penalty before the data were analysed. They indicated their informed consent by signing the consent form.

## Exploratory stage

### Sample and data collection procedure

Interviews were carried out with a total of 18 SNS users so that a more detailed understanding of the key information privacy factors that impact on SNS users in terms of information sharing could be obtained. Users' information sharing habits across SNSs were discussed in detail in the interviews. Each participant was an academic, an experienced SNS user with between one and five years' experience of regular information sharing across sites; participants were also diverse in their discipline as well as cultural background, to include wider views about the research issue. Table 1 provides interviewees' characteristic information. Further details of sampling and procedure can be found in Mutambik et al. [51], along with a focus on the characteristic of gender, which we do not further address here except in relation to culture.

**Table 1. Participants' characteristic distribution.**

| Participant Characteristic | | Frequency |
|---|---|---|
| Gender | Male | 10 |
| | Female | 8 |
| SNS experience | > 3 year | 7 |
| | 3 to 5 | 6 |
| | < 5 | 5 |
| Culture | Arab | 12 |
| | Non-Arab | 6 |
| Academic level | Student | 11 |
| | Lecturer | 7 |

The following section gives the results of the exploratory stage, thus setting the basis for the model and the hypothesis for the research.

## Results of the exploratory study

The findings of the exploratory study stage were reached using a thematic analysis approach, which is "a method for identifying, analysing and reporting patterns (themes) within data" [53, p.57]. In this approach, the researcher draws themes from the data and examination and interpretation of these themes can help to determine how the research question can be answered. The rigour of this particular process was ensured by triangulating the data so that any agreement among participants (convergence) and any disagreements (divergence) could be identified. In order to accomplish this, the responses of all participants were compared according to demographics–in particular, Arab vs. non-Arab.

The findings showed three information privacy concern factors that were considered by the interviewees to influence SNS users' decision to share information, namely: Personal Information Collection & Usage (PICU), Personal Information Control (PIC), and Awareness of Effectiveness of Privacy Policy (AEPP). These factors are defined as follows:

**PICU**–the degree to which a user is concerned about the risk presented by individual-specific information being collected and used by SNS;

**PIC**–the degree to which a user believes they have control over their individual-specific information when using SNS;

**AEPP**–the degree to which a user believes that SNS privacy statements are important to understanding and accepting the safety of their information.

Table 2 provides examples of respondents' statements, illustrating the findings from the exploratory study. We include these in more detail than Mutambik et al. [51] because we will focus especially on details of attitudes towards privacy concerns.

## Developing the research model and hypotheses

When an individual provides personal information to SNS, a long-term exchange relationship is begun in the context of information privacy. Thus, the research model was developed first from informal discussions of the findings from the initial stage, then through comparing those findings with prior literature in the area of information sharing, social networking and internet privacy. We note that SNS users in the present study highlighted the importance of PICU, which affects their decision to use SNS; an appreciation of the risks is valuable, but if they have too high a level of PICU, they may be unsuccessful in their attempts to use SNS as the information sharing practices used may seem dangerous. A high level of PIC is also needed, as this is how new abilities are developed using SNS and subsequently affects their decision to share information. Moreover, it was found that a high level of AEPP has an impact on users' intention to share information, which then influences SNS users' information sharing as well. We note that the relationships here are complex: AEPP has a significant bearing on PICU, and perhaps it gets much of its influence on attitude and intention via PICU and PIC.

Our focus is on the potential influence of national culture, and we hypothesise that this influence on behavioural intentions will be notably evident in users' *attitudes* towards privacy and sharing. Here we diverge from Mutambik et al. [51], where attitudes, although regularly mentioned, do not form an explicit part of the model. Our model, diagrammed in Fig 1, presents these factors based on the theory of reasoned action (TRA) [10], and proposes a more detailed view of how PICU, PIC and AEPP influence a user's attitudes as well as intentions, and ultimately behaviour, in using SNS.

**Table 2. Interviews' main findings with respondent statements as examples.**

| Factors | Respondent statements |
|---|---|
| PICU | "As a user of social networking sites, I need to understand clearly what is fair and what is not fair regarding what these sites are collecting and how they are using my personal information. . . . When I share my personal information or these sites gather it, I need to understand the privacy aspect in this regard and social networking companies should take into account the security of the information." |
| | "Privacy of any information I share with social sites is important . . . gathering personally identifiable data from me or other users is OK, but we need to be informed about intended use of the information. . . . When sites gather too much personal data it sometimes makes me hesitant to use them, but nowadays we need to use them and the risk is that our data is being with other companies without our knowledge." |
| | "Gathering personal data really affects my decision to use social media sites but, you know, this is the central theme of information sharing often based on the agreed social contract that we should read." |
| | "The way data is collected, legally or illegally, is the main point of information privacy concerns when sharing information with social media sites and can sometimes stop people from using them . . . I think social media sites nowadays make promises to save personal data provided by their users safely, trying to ensure the privacy aspect." |
| | "Actually, we give up some information in return for the value that social media sites offer us. I am reluctant to share my personal information if I expect it to be used negatively. This is an important factor that will affect my decision to share my information." |
| PIC | "I think the social networking sites give us different ways to control our personal information which is often clearly set out in their terms and conditions. With the widespread adoption and use of social media sites, what I think is more important is having the skills and knowledge to control what personal information to share with them." |
| | "What I need when sharing information in general, and personal information when using social media sites, is an understanding of contractual terms and the knowledge that I have self-control over it during the relationship and have the freedom to exit when I want to." |
| | "Users' concerns about privacy of personal information manifests in whether the individual has control over personal information, such as giving approval to use it, modification when necessary, and the option to opt in/out. Clarity on these factors will certainly affect information sharing on social networking." |
| | "When I am guaranteed control over my information and I am informed continuously about the intended use of my information, which sometimes social media sites do, I'm happy to share my information." |
| AEPP | "Before I agree to the terms and conditions of any social media site, I read the privacy policies of the information I share with them . . . privacy policies matter to me so I can decide which information to share with these sites. This is not a general case, as not all users read privacy policies . . . The main point I want to make here is that privacy policies are often built to suit the laws in western countries, and not others . . . I think such policies should be clear and effective and customised for each country . . . culture plays an important role here." |
| | "Users of social media sites should be aware of the information sharing practices used by each site . . . what they collect, how they control it, what they decide to share, practices they use to protect personal data, and with whom they share it . . . all of these should be outlined in clear privacy policies when we sign up to use social sites. If this is available to users, it shows that the sites have effective policies and take information privacy seriously, making users aware of their rights." |
| | "I use social media sites when I believe that my information is protected and their privacy policies are reliable. I also need to be aware and appropriately informed about the use of information sharing policies." |

The next sections discuss in more detail the influence of the aforementioned factors and how hypotheses can be derived from them, which will then be used in the next stage of the study. This adds substantial detail compared with the set of hypotheses investigated by Mutambik et al. [51]. The model itself does not imply cultural factors as a variable per se, however we propose that differential effects are revealed when we analyse how each of the factors combine using culture as a lens and moderating variable that operates across the links.

**The influence of PIC on SNS for information sharing.** The participants mentioned the need for control over the information that SNS users share on the sites, as there is a strong connection between perceived control and information sharing behaviours and decision making. The identification and inclusion of perceived control in this research echoes similar concerns in existing literature. For example, from a psychological perspective, researchers have suggested

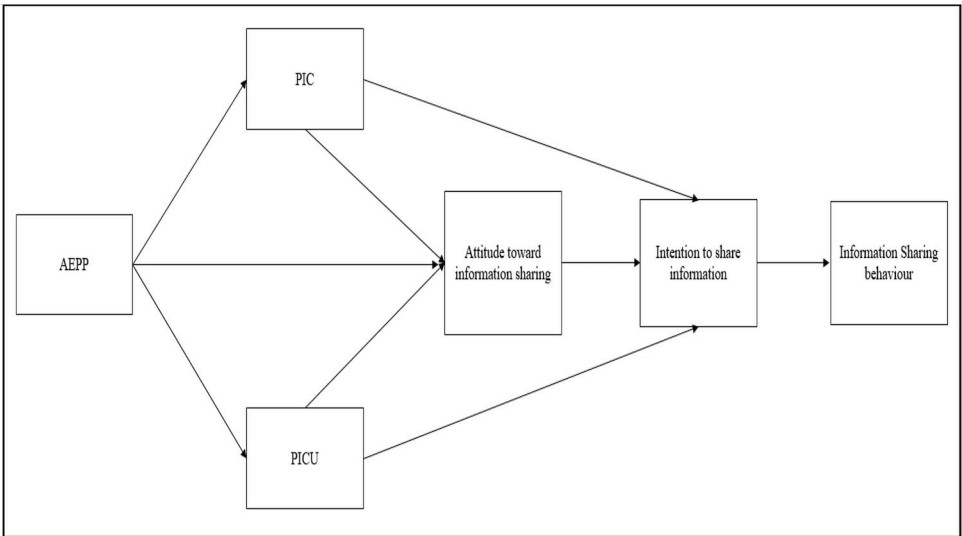

**Fig 1. Research model.**

that individuals' behaviours are influenced by their perceived control [54–56]. Moreover, if a person has perceived control over a situation then certain outcomes are achieved that may facilitate information sharing, such as user acceptance of technology [57, 58]. This kind of psychological perception of control increases the likelihood of a certain behaviour.

Further, when reviewing internet privacy and personal information literature in general, perceived control plays a significant role in information privacy online, and there is no reason for this to be any different for SNS. For example, researchers [59–61] have shown that a person's attitude towards sharing data online is heavily influenced by their perceived control. Decision making by users of Facebook and Twitter, for example, is based on how much control users feel they have over the use of their information [62]. Therefore, it could be argued that an increased perception of control leads to more positive attitudes towards information sharing, and thus SNS users will feel more confident sharing their personal details with fewer concerns about questionable data collection (although there is then, of course, some risk of over-confidence). To explore the influence of perceived control in the context of personal information sharing on SNS, the following hypothesis is advanced:

**H1**. PIC influences users' attitudes toward using SNS for information sharing.

As evidenced by Hoadley et al. [63], Facebook users perceive themselves to have less control over private information due to the 'newsfeed' feature; it provides easier access to information which means that users assume that there is a greater risk of privacy intrusion, resulting in them being less likely to share their information on SNS platforms. If users perceive their control over information to be high, their privacy concerns are likely to be reduced and their perception of the security of using SNS will improve [60, 64]. Hence, the higher an SNS user's perceived control regarding their personal information, the higher is their intention to share such information on SNS platforms. With this in mind, it can therefore be argued that users' intention to share information on SNS is impacted significantly by level of perceived control. To be more specific, SNS users are more likely to share personal information on the platforms if they perceive themselves to have greater control over that information. To understand this in the area of personal information sharing on SNS, the following hypothesis is proposed:

**H2**. PIC influences users' likelihood to use SNS for information sharing.

*The influence of AEPP on SNS for information sharing.* Contrary to existing research, the responses from this study's interviewees (as presented in Table 2) illustrate the importance of privacy policies are in influencing the information sharing behaviours of SNS users. In accordance with prior studies, users' awareness of privacy policies and their effectiveness revolves around the risk of unauthorised use of personal data [65, 66]. It is known that users' attitudes are impacted by this type of privacy policy awareness, including users' willingness to share information [64, 67] and make transactions online [68, 69]. Likewise, an individual's lack of perception of the effectiveness of privacy policies has a negative impact on their likelihood of providing personal information online [70]. In principle, SNS users' data could be obtained, shared, and used very easily without the user knowing or consenting at all. Knowing this may lead users to perceive these sites as risky (compromising PICU and PIC), resulting in an unwillingness to share data. With this in mind, users will develop a more positive attitude towards sharing information online if they are confident that their data will be treated carefully and appropriately. Therefore, based on the extant literature in this field, along with the study's interviews and the conceptualisation process, the following three hypotheses were developed:

**H3**. AEPP influences users' attitudes toward using SNS for information sharing.

**H4**. AEPP influences users' PIC to use SNS for information sharing.

**H5**. AEPP influences users' PICU to use SNS for information sharing.

*The influence of PICU on SNS for information sharing.* One of the main concerns people have about sharing their information with websites is related to data collection and use, in particular when making transactions [61, 71]. Concerns related to data collection are one of the key factors that impact the use of online services [72], and SNS are no exception to this. Looking at Table 2, for instance, the findings from the exploratory stage of the study show that the participants found SNS data collection to be the primary cause for concern when sharing their information with the websites. Previous literature also discusses these concerns–for example, Gillespie and Dietz [73] demonstrated that a person's decision to utilise an online service is significantly affected by their perception of the website's approach to data collection. Ayaburi and Treku [74] also showed that individuals' attitudes are affected by concerns over data collection on SNS; however, users can rebuild their trust in a site as long as there is an appropriate response, diagnosis, intervention and assessment made by the site in question and efforts are made to avoid any future privacy breaches [73, 74]. Consistent with the findings from the exploratory study and literature review regarding internet privacy concerns, we hypothesise that:

**H6**. PICU influences users' attitudes toward using SNS for information sharing.

**H7**. PICU influences users' intention to use SNS for information sharing.

*The influence of attitude and intention on SNS behaviour for information sharing.* Originating in discourse on the philosophy of free will, TRA seeks to link attitudes and behaviour or intentions and actions: to what extent is the individual in control of her destiny and to what degree is she the creature of external forces? Developed in its current form by Fishbein and Ajzen in 1975, TRA is often cited in social policy and marketing studies to explain why policies or products gain or fail to gain acceptance. As Al-Harbi [75] notes, TRA focuses on the individual's motivation in the sense of her/his conscious plan or decision to engage in the behaviour i.e. how can the steps between intention and action be decomposed and reinforced. For rational justification of and emotional attachment to intention are key to understanding when intention results in behaviour. As Ajzen and Fishbein [76, p. 6] note, attitude is the individual's

positive or negative evaluation of performing the behaviour. In the context of the present research, individuals will be more likely to share information if they have a greater intention to share information on SNS than others do. Therefore, we hypothesise that:

**H8.** Attitudes toward using SNS influences users' intention to share information using SNS.

**H9.** Intention to share information influences users' information sharing behaviour on SNS.

*The role of gender.* The role of gender is, as noted above, covered in detail by Mutambik et al. [51], so we do not discuss it further here, except to note that clear gender differences are shown there to emerge, and that, although it seems superfluous to present the evidence in detail, we have found these differences in attitude as well as intention and behaviour.

*The role of culture.* As discussed above, a central intention of our study is to investigate the possible existence of a cultural distinction in responses to issues concerning information sharing between Arab and non-Arab members of our population. We conjecture that such differences will show up in potentially all of the relationships diagrammed in Fig 1. Hence, we hypothesise:

**H10**. The PIC influences on users' attitudes toward using SNS for information sharing differ between Arab and Non-Arab users.

**H11**. The PIC influences on users' intention to use SNS for information sharing differ between Arab and Non-Arab users.

**H12**. The AEPP influences on users' attitudes toward using SNS for information sharing differ between Arab and Non-Arab users.

**H13**. The AEPP influences on users' PIC to use SNS for information sharing differ between Arab and Non-Arab users.

**H14**. The AEPP influences on users' PICU to use SNS for information sharing differ between Arab and Non-Arab users.

**H15**. The PICU influences on users' attitudes toward using SNS for information sharing differ between Arab and Non-Arab users.

**H16**. The PICU influences on users' intention to use SNS for information sharing differ between Arab and Non-Arab users.

**H17.** The influence of attitudes toward using SNS on users' intentions to share information using SNS differs between Arab and Non-Arab users.

**H18.** The intention to share information differs between Arab and Non-Arab users in its influence on users' information sharing behaviour on SNS.

## Confirmatory stage

As soon as the research model was formed, as explained above, the second stage was conducted. The next sub-section presents a description of how the questionnaire was developed, how the validated data were collected, and the sample of this stage in order to validate the study's model and hypotheses. This is in common with Mutambik et al. [51], and further details may be found there of the development of the questionnaire, the content validity assessment and collection method, which we omit here for reasons of space. We retain Table 4, below, for coherence of the statistical account given that the version here adds the attitude construct (ATT).

Table 3 summarises the key demographics of the participants.

**Table 3. Participant characteristic distribution for the questionnaire.**

| Participant Characteristic | | Frequency |
|---|---|---|
| Gender | Male | 221 |
| | Female | 191 |
| SNS experience | > 3 year | 112 |
| | 3 to 5 | 203 |
| | < 5 | 97 |
| Culture | Arab | 302 |
| | Non-Arab | 110 |
| Academic level | Student | 128 |
| | Lecture | 224 |
| | Administration staff | 60 |

## Data analysis techniques

The Statistical Package for the Social Sciences (SPSS) version 21 was used to code and clean up the raw data collected during the confirmatory study stage. For instance, the minimum, maximum, and frequency scores were worked out for each question to ensure the accuracy of each data score. Lastly, as per recommendations by Hair et al. [77], the information was thoroughly examined so that any missed data could be identified. The psychometric properties of the measurement model were analysed using structural equation modelling and the hypothesis tested using the same approach. Estimations were made at this stage using the Amos (v.26) software package.

## Testing measurement model

Testing of the measurement model's strength was first done using construct validity, which is defined by [78] as "the extent to which a measured variable actually measures the conceptual variable (the construct) that it is designed to assess". Following this, internal consistency reliability was assessed, which is whether or not the instrument measures what it is supposed to [79]. Certain criteria are outlined by [77, 80] which are a factor loading greater than 0.6, Cronbach's alpha (CA) and composite reliability (CR) of greater than 0.7, and an average variance extracted (AVE) of over 0.5 –all of which were used in the study for the measurement of model quality.

Table 4 presents the results of the analysis and illustrates that each of the factor loadings meets or exceeds the criterion of 0.6. In addition, as evidenced by Table 5, the CR and CA values exceed the criterion of 0.7, with AVE reaching more than 0.5 for each construct. The bold values seen in Table 5 are the square root values for the AVE of each construct; these values are observably higher than those on the rows below. It can therefore be concluded that both the convergent and discriminant validity, as well as the internal consistency reliability, of the model are confirmed.

## Common method variance

Since self-contained questionnaires were used to collect the data for the study, it is possible that common method variance (CMV) could be introduced which may lead to lowered variable validity–this has the ability to impact on the acceptance or rejection of a hypothesis [81]. CMV can be evaluated using confirmatory factor analysis through the Harman single factor test method [81, 82]. Here, each of the variable items are analysed together using SPSS and a number of factors are extracted (without specifying the rotation). The SPSS running results

**Table 4. Constructs, items with factor loadings, and sources.**

| Constructs | Items | Loading | Source |
|---|---|---|---|
| ATT | ATT1: Sharing information using SNS is a good idea. | 0.89 | [9]. |
| | ATT2: Sharing information using SNS is a wise idea. | 0.86 | |
| | ATT3: I like the idea of sharing information using SNS. | 0.90 | |
| ISI | ISI1: I intend to carry on sharing information using SNS. | 0.86 | [9]. |
| | ISI2. I plan to carry on sharing information using SNS frequently. | 0.89 | |
| | ISI3. I will carry on try to share information using SNS in my daily life. | 0.88 | |
| IS | IS1: I often participate in information/knowledge sharing activities using SNS. | 0.90 | [9]. |
| | IS2: I often share my experience or knowledge with others on SNS. | 0.89 | |
| | IS3: When participating on SNS, I usually actively share my information and knowledge with others. | 0.78 | |
| PICU | PICU1: It usually worries me when SNS ask me for personal information. | 0.78 | Self-develop based on the qualitative data & [33]. |
| | PICU2: When SNS ask me for personal information, I sometimes think twice before providing it. | 0.82 | |
| | PICU3: It worries me to give personal information to so many SNS. | 0.84 | |
| | PICU4: I'm concerned that SNS are using my personal information for their business. | 0.85 | |
| PIC | PIC1: User control of personal information is the heart of user privacy. | 0.78 | Self-develop based on the qualitative data & [58]. |
| | PIC2: I feel in control over the information I provide when using SNS. | 0.75 | |
| | PIC3: Privacy setting give me full control over the information I provide when using SNS. | 0.72 | |
| | PIC4: I feel in control of who can view my information when using SNS. | 0.81 | |
| AEPP | AEPP1: I believe that SNS privacy statements are an effective way to demonstrate their commitments to privacy. | 0.77 | Self-develop based on the qualitative data & [70]. |
| | AEPP2: With their privacy statements, I believe that my personal information will be kept private and confidential by SNS. | 0.71 | |
| | AEPP3: I feel confident that SNS privacy statements reflect their commitments to protect my personal information. | 0.86 | |

indicate that from the extracted factors, the first main component's variance is 32.18% without rotation; this is not accounted for the majority. Hence, CMV is not a major issue and will not impact significantly on the variables' path coefficients.

## Results of structural model evaluation

Structural model analysis and testing of the hypothesis was carried out using the Amos program, for which the results are provided in Fig 2. It can be seen that: PIC, PICU and AEPP explain 39.1% of the variance in attitude towards information sharing; PIC, PICU and AEPP with attitude explain 48.1% of variance in intention to share information; and 66.2% of

**Table 5. Correlations, Cronbach's alpha (CA), composite reliability (CR) and average variance extracted (AVE).**

| Constructs | CA | CR | AVE | Correlations | | | | | |
|---|---|---|---|---|---|---|---|---|---|
| | | | | ATT | ISI | IS | PICU | PIC | AEPP |
| ATT | 0.92 | 0.91 | 0.80 | **0.89** | | | | | |
| ISI | 0.89 | 0.91 | 0.78 | 0.61 | **0.88** | | | | |
| IS | 0.91 | 0.89 | 0.75 | 0.66 | 0.52 | **0.87** | | | |
| PICU | 0.88 | 0.91 | 0.69 | 0.70 | 0.68 | 0.76 | **0.83** | | |
| PIC | 0.91 | 0.88 | 0.58 | 0.52 | 0.48 | 0.65 | 0.75 | **0.76** | |
| AEPP | 0.92 | 0.82 | 0.60 | 0.39 | 0.36 | 0.62 | 0.65 | 0.55 | **0.77** |

Note: Square root of AVE shown in bold as the diagonal. Notes: ATT: Attitude; ISI: Intention to share information; IS: Information Sharing; PICU: Personal Information Collection & Usage; PIC: Personal Information Control; AEPP: Awareness of Effectiveness of Privacy Policy

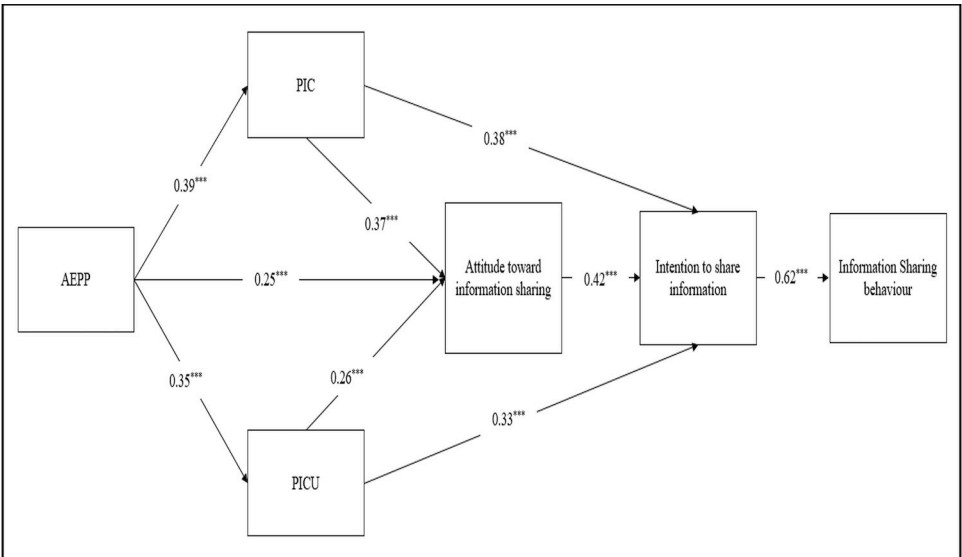

**Fig 2. Results of structural model evaluation using the whole sample.** *** p < 0.01.

variance in information sharing behaviour is explained by intention to share information–suggesting that a substantial amount of variance is explained [83]. The study findings also indicate that all hypotheses are fully supported. The results show that PICU, PIC, and AEPP all have significant influence over individuals' attitudes towards online information sharing behaviours, which subsequently affects users' information sharing behaviours on SNS. The *t*-values and standardised path coefficients of the model are presented in detail in Table 6.

## Analysis of the model paths for comparison of Arab vs. non-Arab

Since it is covered by Mutambik et al. [51], we do not give an analysis of gender comparison. Group differences between the Arab and non-Arab groups were analysed using a multigroup partial least squares (PLS) analysis, which was done by comparing the corresponding path coefficients in both groups' structural model. This subsection explains and presents the findings of this comparison in more detail.

As shown in Table 3, the full dataset comprised 412 participants (302 Arab and 110 non-Arab participants). Based on this distribution, the analysis in Table 7 shows that the

**Table 6. Path coefficients and t-values for the whole sample.**

| Hypothesis | Standardised path coefficient | *t*-value | Support? |
|---|---|---|---|
| H1: PIC ➔ attitudes toward using SNS for information sharing | 0.37 | 5.2!*** | YES |
| H2: PIC ➔ intention to use SNS for information sharing | 0.40 | 5.29*** | YES |
| H3: AEPP ➔ attitudes toward using SNS for information sharing | 0.25 | 4.71*** | YES |
| H4: AEPP ➔ PIC | 0.39 | 5.26*** | YES |
| H5: AEPP ➔ PICU | 0.35 | 5.21*** | YES |
| H6: PICU ➔ attitudes toward using SNS for information sharing | 0.26 | 4.88*** | YES |
| H7: PICU ➔ intention to use SNS for information sharing | 0.38 | 5.51*** | YES |
| H8: attitudes toward using SNS ➔ intention to use SNS for information sharing | 0.42 | 5.88*** | YES |
| H9: intention to use SNS for information sharing ➔ information sharing behaviour | 0.67 | 7.25*** | YES |

Note: ***: 0.001 significance

standardised path coefficients for the Arab group are higher compared to the non-Arab group in regard to all of the relationships tested. Notwithstanding the difference in group size, these results supported H10-18. Hence, differences do exist between these groups in this context, and we assert that cultural factors lie behind these differences.

## Discussion

In the modern era of information technology, one of the main ethical considerations to contend with is the appropriate and transparent use of people's personal information. Consequently, online privacy has been a primary topic of many marketing and e-commerce studies [84–86], and has touched on SNS in a limited way, especially in recent years [2, 87]. Studies regarding online privacy tend to be concerned with the moral and ethical behaviours of people, firms, and organisations. In the present study, however, we reveal privacy issues raised by users' concerns about protection of their information on SNS as a tool for information sharing. This means that this research makes a contribution to the literature regarding privacy concerns, but also extends it to include users' perspectives.

Unlike many previous studies [2, 65, 84, 87, 88], the current study utilised detailed, face-to-face interviews along with the most relevant existing literature to create a model, in a systematic manner, for use in explaining information sharing concerns from the perspective of SNS users specifically. The study therefore contributes a more detailed understanding of individuals and their decision to share personal data on SNS, as well as illuminating the important influences of PIC, PICU, and AEPP in this context.

For instance, the empirical findings revealed that the attitudes and intention of SNS users are strongly influenced by PICU–the individual's perception of risk–which subsequently affects their data sharing behaviours. In line with Jozani et al. [37], this implies that although SNS platforms want to gather as much information as possible from users, the privacy of those users must be respected and protected, and this must be recognised by users effectively to reduce the perception of risk.

Moreover, the results demonstrated that the attitudes and intentions of SNS users are also strongly influenced by PIC–the individual's perception of control over their shared information–which, again, has an impact on their data sharing behaviours. The findings show the importance of users' perceived control of their shared information–much more than actual

**Table 7. Standardised comparisons of paths between Arab and non-Arab.**

| Hypothesis | Arab | | Non-Arab | | Standardised comparisons of paths | Support? |
|---|---|---|---|---|---|---|
| | (*n* = 302) | | (*n* = 110) | | | |
| | Standardised path coefficient | t-value | Standardised path coefficient | t-value | Δ path (Arab–Non-Arab) | |
| H10: PIC ➔ attitudes toward using SNS | 0.47*** | 5.92 | 0.32** | 3.95 | 0.15 | YES |
| H11: PIC ➔ intention to use SNS | 0.28*** | 4.31 | 0.08 | 0.82 | 0.2 | YES |
| H12: AEPP ➔ attitudes to use SNS | 0.38*** | 3.09 | 0.23** | 2.89 | 0.15 | YES |
| H13: AEPP ➔ PIC to use SNS | 0.48*** | 3.6 | 0.27** | 2.66 | 0.21 | YES |
| H14: AEPP ➔ PICU to use SNS | 0.49*** | 3.10 | 0.29** | 2.59 | 0.20 | YES |
| H15: PICU ➔ intention to use SNS | 0.7*** | 4.67 | 0.49*** | 3.97 | 0.21 | YES |
| H16: PICU ➔ intention to use SNS | 0.33*** | 3.99 | 0.10 | 0.98 | 0.23 | YES |
| H17: attitudes toward using SNS ➔ intention to use SNS for information sharing | 0.58*** | 6.59 | 0.31*** | 4.02 | 0.27 | YES |
| H18: intention to use SNS for information sharing ➔ information sharing behaviour | 0.49*** | 5.03 | 0.27** | 2.01 | 0.22 | YES |

control of information–in terms of increasing perceived privacy on social networking platforms. Users of SNS are encouraged to share a significant amount of personal information, or information from which personal information can be inferred, which poses quite considerable challenges to keeping users' information safe. As such, in order to increase users' trust, SNS are given the opportunity to more effectively manage privacy risks through obtaining valuable insights into how PIC impacts on users' information sharing behaviours. For example, Berings and Adriaenssens [89] suggest that it is possible to reduce an individual's perceived privacy risk by offering them a range of options for controlling their data, such as more practical privacy settings. This way, information sharing on SNS platforms becomes more sustainable because users will have greater PIC. Organisations and businesses are also then provided with the opportunity to increase their values. On the whole, SNS users want greater levels of privacy on these sites, as reflected in the ability to better control the use of the content they post there.

Central to the notion of privacy is users' control over their own private information. This is a moral and ethical right, and one which must be upheld by SNS if they are to be trusted with users' information. The particular importance of this to users in the Arab world is emphasised by Abokhodair et al. [50]. SNS users share content with the expectation that it will only be seen by the people they choose to share it with; however, even if this is the case, other data may be collected in relation to this content, others' interactions with it and the user's interactions with other data, so that highly personal information may nonetheless be shared with other businesses and individuals. If PIC is higher than actual control, then sharing is not as optimal. Consequently, more data is able to be shared in unexpected ways by SNS and privacy is invaded. The current research suggests that companies must strike more of a balance between users' PIC and actual control through the use of more adaptable privacy settings. In that case, users are able to better manage their own information privacy and therefore obtain greater control over it. However, there will be a trade-off between complexity and doing what the user wants, such that simplifying controls may make them more usable, but less detailed in the ways the user can control things. Mourey and Waldman [90] intriguingly suggest that users' estimation of the importance of privacy itself varies depending on the degree of control users feel that they have, implying that the interplay of factors in this area may be more complex than previously thought.

Many of our interviewees seem convinced that privacy policies used by SNS providers are important in informing users of SNS systems, and sometimes that if users knew them better then they might be less fearful about sharing information.

> *I am aware that all the big companies collect user's data from SNS and that these companies have long advocated regulating privacy and data protection through self-regulation. . . These self-regulatory mechanisms provide greater flexibility for users . . . But many users still struggle because of the technicalities and/or lengthy nature of these policies.*

> *For me, before I agree to the terms and conditions of any of the SNS, I ensure that I read the privacy policies that cover the information I share with them . . . The privacy policies matter to me and affect the information I choose to share.*

> *SNS users do not have sufficient awareness of privacy policies and how effective they are. For example, how does a user seek redress when something goes wrong and their information safety is compromised?*

> *Information privacy is radically different from traditional privacy methods. In social networks there is not enough awareness of the effectiveness of privacy policies, which may lead to hesitancy . . . As users of social networks for the purpose of sharing knowledge, we need to be aware of the policies that are supposed to guarantee protection of our personal data.*

There are also claims in the literature that privacy policies are important to users of many types of information systems, whether SNS, banking, retail, mobile etc. For example, it is said that "privacy notices have become an important means for reducing consumers' privacy concerns by providing them with information about how companies use the collected data" [91, p. 889].

This motivates the following more detailed discussion of the practical importance of privacy policies in general and what may influence users' attitudes towards them. In particular, we try to identify aspects of the general situation that may be helping to drive differences in attitude to the importance of privacy policies, in ways that may relate to national culture.

It's a commonplace of everyday experience that people very rarely read these policies when signing up to a new web site or mobile app–certainly not in any detail, and very often not at all. Even when they do read them, they are often hard to understand. Some of our interviewees also recognise this, e.g.:

*Awareness of SNS privacy policies*? *My goodness*! *Who on earth knows these things*, *except lawyers who work for these companies*? *Have you ever tried reading their privacy policy statements*?

A number of previous studies highlight the importance of effective privacy policies; however, it is worth noting that, on the whole, they find that the existence of a privacy policy is more important than its content. Sometimes this is very explicit, e.g., "An e-store that fails to include a privacy policy will lose consumers' trust, but the impact of privacy disclosures does not extend much beyond this point . . . shoppers do not typically read the policy" [92]. More often, phrases are used such as perceived "effectiveness of privacy policies" [66], and it turns out on closer inspection of the methodology that these dimensions do not depend on any knowledge of the content of the policies. A review by Mitchell and El-Gayar [7, p. 4228] concludes that "Privacy policies do not have a direct effect on the information sharing behaviour of OSN users" and that "users generally do not read these sometimes-laborious policies".

A recent and especially trenchant rejection of the idea that users read policy statements comes from Obar and Oeldorf-Hirsch [41, p. 129], who address "the biggest lie on the internet", which is *I have read and agree to the terms*. They argue convincingly that reading the Terms of Service (TOS, generally including the privacy statement), where reported, is often over-reported; they conduct a study showing that when people were asked to sign up to an SNS service, only 26% even looked at the privacy policy, and even then mostly very briefly. Both qualitative and quantitative results suggested that users experience or anticipate "information overload" and that "Privacy and TOS policies were seen as more of a nuisance than anything else" [41, p. 142]. They identify this as a serious problem, since it clearly means that users are not engaging with informed consent processes.

Most regulatory systems, notably the European Union's General Data Protection Regulations [93], impose an obligation to obtain "informed consent" from users concerning the handling of their data. Yet users seem typically willing to forego being informed. Is this rational? Our interviewees often suggest that users need more awareness of privacy policies, not less; our AEPP results, and data across the literature, suggest that people regard the content of policies as important, yet they are not concerned to access them. This may be presented as an aspect of the "privacy paradox" [94, 95]. However, perhaps one has to ask the question: what would be the utility of detailed knowledge of privacy policies? What could the user do with it? Would it be likely to lead the user to be less (or more) fearful or cautious?

In practice, SNS and other systems provide various controls for users to restrict access to their data, especially by other users. Users will often be concerned to use these controls effectively, as

indicated by the importance of PIC, but the privacy policy provides little advice on doing that. There are other types of data, e.g. a user's "likes", which are usually not controllable. Privacy policies may indicate how such data may be used, although still they cannot be controlled. Despite the length and complexity of the policies, they do not in general contain fine detail on the types of data collected and the collection methods, nor on their precise uses. The Facebook policy, for instance, says rather vaguely that "We collect information about how you use our Products, such as the types of content that you view or engage with, the features you use, the actions you take, the people or accounts you interact with and the time, frequency and duration of your activities" [96]: in other words, more or less anything they could possibly collect may be collected, and subsequently it's implied that any of this that can legally be shared may be shared with "partners".

The regulatory framework around policies is often noted in the literature as a major influence on their content and usage. Many privacy policies are built around the US Federal Trade Commission's Fair Information Practice Principles, the US practice being also followed in many other countries [66, 70]. Clearly many [96] are also influenced by the GDPR [93] and related European regulations. This being the case, users in such jurisdictions may tend to feel that their interests are protected by the regulations, and that the main role of the privacy policy is to acknowledge that the provider is aware of the need to comply with these.

Obar and Oeldorf-Hirsch [41] note that TOS in general may commit the user to things they would not accept if they knew about them; rather extremely, they present their users with a TOS that agrees to give the service provider their first-born child, and almost all their users agree to this without noticing it. However, it seems clear that such a clause would not be enforceable, being contrary to a widespread understanding of principles of "good faith" [97]. In general, terms relating to data privacy will have to be *reasonable* and will have to comply with the regulations.

Consider, then, the following line of thought. Any commitment in a privacy statement that is legally enforceable is probably legally required. An SNS provider will probably not commit themselves, in their TOS, to a provision that is not required of them by law, and even if they do, the typical user will never attempt to hold them to it. It follows that, in a strong regulatory jurisdiction, the privacy statement will not add anything practically useful to the provisions of the regulations.

A policy may allow things that the user would find unacceptable, even though they are "reasonable", and perhaps there are things (such as giving information to the government) that the provider can be obliged to do regardless of the policy. In any case, the user's sole alternative to accepting these terms is to decline the service. It is clear that SNS users, especially of large and popular services, commonly feel that the benefits outweigh any risks that might plausibly exist within or around the privacy statement.

Suspicion that the terms (and even the regulations) do not rule out dubious practices may well drive the user's behaviour in the direction of being more cautious about disclosing information, but it seems unlikely that deeper knowledge of the privacy policy would encourage, or justify, any greater boldness. Boldness is perhaps only likely to arise from an understanding of the controls available, which in an SNS mostly apply to data/information directly shared for the consumption of other users.

We included, above, the caveat "in a strong regulatory jurisdiction": if regulation is not strong, then the position may differ, in that users will not have the same degree of expectation that providers' behaviour is constrained and predictable. It is commonly noted that consumer confidence in general is somewhat dependent on regulatory frameworks [98]. In Saudi Arabia, at the time of our data collection, there was and is no specific legislation for data protection [99]. In a context like this, users of information systems should be expected to show greater caution and concern. Privacy policies may be felt to offer some sort of contractual reassurance

that providers will adopt reasonable practices, and hence there is some reason to anticipate users' gaining confidence through increased knowledge of a policy's provisions. Bellman et al. [29] studied attitudes to privacy across 38 countries and found that participants from countries lacking privacy regulation were more concerned about the security of online transactions than those from countries with regulation. Several of our interviewees are especially concerned that privacy policies are not drawn up to respect the Arab cultural context, and seem to feel that this implies further, if unclear and perhaps complex risks to negotiate.

> *The main concern for me as an Arab female is that such privacy policies are generic and often built to fit the laws of western countries. I would love it if they were country- or context-specific so that they cater to the needs of my culture and religion.*

> *The privacy policies matter to me to determine what information I share with these sites. This is not a general case, as not all users read the privacy policies . . . the main point is that privacy policies are often built to fit the laws of western countries, and not others . . . I think such policies should be clear and effective and customised for each country . . . Culture plays an important role here.*

> *Social networking sites deal with users from different cultures . . . Cultural differences, although useful in some cases, represent some challenges for social media, and they also make it difficult for some individuals to develop a sense of trust, because all cultures are different.*

> *A point that caught my attention is that what applies to the European Union may not apply to other countries, so should social media sites adapt and harmonise the laws and terms of use in proportion to each country? This is very important, as each country has its own laws and its own culture, and what applies to the European Union may not apply to other countries, so I stress the importance of adapting and harmonising the laws and terms of use. This will reinforce a more positive attitude.*

Relatedly, an important consideration is whether and how one can obtain redress if there is a problem arising from a privacy issue.

> *I often think that none of these companies are in the Arab world. What does one do when there is a misuse of one's data? There are no clear cut answers to issues like that in any of the privacy policies. We also know, or least hear about, how powerful these companies have become over the years. It is scary to even imagine taking any of them to court if there has been a breach of information.*

> *SNS users do not have sufficient awareness of their privacy policies and how effective these policies are. For example, how does a user seek redress when something goes wrong and information safety is compromised? Most SNS users don't know the answer to this question. So, for me, it is not as safe a medium as it looks.*

In the absence of legislation that explicitly provides for such issues, the route is likely to be through more general contract or consumer protection law. For that to be effective, the specific nature of the contract, hence the TOS, is critical to users' feeling comfortable that their interests are protected.

We conjecture that the regulatory context is a significant influence on our interviewees' attitudes to privacy policies. However, we recall also that the interviewees are not culturally homogeneous. All work within the Saudi system, but some are members of the majority Saudi culture (which we designate as *Arab*), while others are not (*non-Arab*). These come from a

variety of other countries and cultures. They do not fully share the Saudi Arab culture, which is a relatively close and homogeneous culture. We have noted in section 2 above that research has exposed differences in concern about privacy between people from different cultures. This may be expected to impose a further level of modulation on the views of our interviewees.

## Conclusion

This study has explored user concerns regarding the sharing of personal information on SNS platforms, with particular reference to users in the Arab world. We have noted widespread general confusion about the role and importance of privacy policies, and see a need for this to be reassessed; however, we have also noted that, relative to results from the literature, which addresses mainly western contexts, our participants may have an enhanced level of concern about privacy policies. We have argued that attitudes to privacy policies are highly subject to the regulatory environment, which in Saudi Arabia is weak, as well as to other culturally specific concerns. The question of whether there is accurate representation of users' ethical concerns is emphasised: from the Arab, and more generally the Muslim perspective, the sharing of personal information, or information from which any personal details can be deduced, intersects issues of individual conduct and relationships that normally demand subtle negotiation. This is part of the fabric of the culture, that can be alarmingly undermined by unexpected and unwarranted disclosures. The relatively crude and opaque controls offered by SNS systems are easily seen as deeply inadequate to these concerns. Here we are building on some of the insights of researchers such as Abokhodair et al. [50]. Future research may usefully develop further the cross-cultural study of ethical issues connected with online user behaviour, which will undoubtedly lead to valuable implications for practice.

## Supporting information

**S1 File. Inclusivity in global research.**
(DOCX)

## Acknowledgments

The authors extend their appreciation to a Researchers Supporting Project at King Saud University, Riyadh, Saudi Arabia.

## Author Contributions

**Conceptualization:** Ibrahim Mutambik, John Lee, Abdullah Almuqrin.

**Data curation:** Ibrahim Mutambik, John Lee, Abdullah Almuqrin.

**Formal analysis:** Ibrahim Mutambik, John Lee, Abdullah Almuqrin, Waleed Halboob, Taha Omar, Ahmad Floos.

**Funding acquisition:** Ibrahim Mutambik, John Lee, Abdullah Almuqrin.

**Investigation:** Ibrahim Mutambik, John Lee, Abdullah Almuqrin.

**Methodology:** Ibrahim Mutambik, John Lee, Abdullah Almuqrin.

**Project administration:** Ibrahim Mutambik, John Lee, Abdullah Almuqrin.

**Resources:** Ibrahim Mutambik, John Lee, Abdullah Almuqrin.

**Software:** Ibrahim Mutambik, John Lee, Abdullah Almuqrin, Waleed Halboob.

**Supervision:** Ibrahim Mutambik, John Lee, Abdullah Almuqrin.

**Validation:** Ibrahim Mutambik, John Lee, Abdullah Almuqrin, Waleed Halboob, Taha Omar, Ahmad Floos.

**Visualization:** Ibrahim Mutambik, John Lee, Abdullah Almuqrin, Taha Omar, Ahmad Floos.

**Writing – original draft:** Ibrahim Mutambik, John Lee, Abdullah Almuqrin.

**Writing – review & editing:** Ibrahim Mutambik, John Lee, Abdullah Almuqrin, Waleed Halboob, Taha Omar, Ahmad Floos.

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
