## [Decision Letter · Decision Letter 0]

13 Dec 2021

PONE-D-21-28070

User concerns regarding information sharing on social networking sites: The user’s perspective

PLOS ONE

Dear Dr. Lee

Thank you for submitting your manuscript to PLOS ONE. After careful consideration, we feel that it has merit but does not fully meet PLOS ONE’s publication criteria as it currently stands. Therefore, we invite you to submit a revised version of the manuscript that addresses the points raised during the review process.

We look forward to receiving your revised manuscript.

Kind regards,

Luigi Lavorgna

Academic Editor

PLOS ONE

“The authors extend their appreciation to Researchers Supporting Project number (RSP-2021/233), King Saud University, Riyadh, Saudi Arabia’

“The authors extend their appreciation to Researchers Supporting Project number (RSP-2021/233), King Saud University, Riyadh, Saudi Arabia”

“The authors extend their appreciation to Researchers Supporting Project number (RSP-2021/233), King Saud University, Riyadh, Saudi Arabia”

5. Please note that in order to use the direct billing option the corresponding author must be affiliated with the chosen institute. Please either amend your manuscript to change the affiliation or corresponding author, or email us at plosone@plos.org with a request to remove this option.

6. Please amend your list of authors on the manuscript to ensure that each author is linked to an affiliation. Authors’ affiliations should reflect the institution where the work was done (if authors moved subsequently, you can also list the new affiliation stating “current affiliation:….” as necessary).

Reviewers' comments:

Reviewer's Responses to Questions

**Comments to the Author**

1. Is the manuscript technically sound, and do the data support the conclusions?

Reviewer #1: No

Reviewer #2: Partly

2. Has the statistical analysis been performed appropriately and rigorously? 

Reviewer #1: No

Reviewer #2: Yes

3. Have the authors made all data underlying the findings in their manuscript fully available?

Reviewer #1: No

Reviewer #2: Yes

4. Is the manuscript presented in an intelligible fashion and written in standard English?

Reviewer #1: No

Reviewer #2: Yes

5. Review Comments to the Author

Reviewer #1: I am seriously concerned about dual publication of results (e.g., DOI: 10.4018/JGIM.2021050110) and on the general structure of the manuscript, with its length and organization suggesting this is a dissertation rather than a research manuscript.

Reviewer #2: Page 3; line 46 : Please give some examples about the application of SNS to professional field. For example in medicine and in neurology there are a lot of implications (Lavorgna L, Lanzillo R, Brescia Morra V, Abbadessa G, Tedeschi G, Bonavita S. Social Media and Multiple Sclerosis in the Posttruth Age. Interact J Med Res. 2017 Sep 27;6(2):e18. doi: 10.2196/ijmr.7879. PMID: 28954717; PMCID: PMC5637067.).

PLease discuss it.

Page 5; line 88: Sharing information is important to raise awereness about different conditions. Can you underline how important is share information in different fields as in economy or medicine. IN neurology for example the Internet is the major source to obtain information about diseases and their treatments ( Moccia M, Brigo F, Tedeschi G, Bonavita S, Lavorgna L. Neurology and the Internet: a review. Neurol Sci. 2018 Jun;39(6):981-987. doi: 10.1007/s10072-018-3339-9. Epub 2018 Mar 28. PMID: 29594831.)

Page 5; line 98: SN was perceived by users to be a useful tool to support health-related coping and social interaction, and may suggest a new kind of therapeutic alliance between physicians and people with chronich deseases in medicine ( Lavorgna L, Russo A, De Stefano M, Lanzillo R, Esposito S, Moshtari F, Rullani F, Piscopo K, Buonanno D, Brescia Morra V, Gallo A, Tedeschi G, Bonavita S. Health-Related Coping and Social Interaction in People with Multiple Sclerosis Supported by a Social Network: Pilot Study With a New Methodological Approach. Interact J Med Res. 2017 Jul 14;6(2):e10. doi: 10.2196/ijmr.7402. PMID: 28710056; PMCID: PMC5533941). Please discuss about SNS are perceived by population.

Page 6; line 113: SN was perceived by users to be a useful tool to support health-related coping and social interaction, and may suggest a new kind of therapeutic alliance between physicians and people with chronich deseases in medicine ( Lavorgna L, Russo A, De Stefano M, Lanzillo R, Esposito S, Moshtari F, Rullani F, Piscopo K, Buonanno D, Brescia Morra V, Gallo A, Tedeschi G, Bonavita S. Health-Related Coping and Social Interaction in People with Multiple Sclerosis Supported by a Social Network: Pilot Study With a New Methodological Approach. Interact J Med Res. 2017 Jul 14;6(2):e10. doi: 10.2196/ijmr.7402. PMID: 28710056; PMCID: PMC5533941). Please discuss about SNS are perceived by population.

Page 50; line 936: It will be appreciate if the authors show also data for other part of the world making a comparison.

6. PLOS authors have the option to publish the peer review history of their article (what does this mean?). If published, this will include your full peer review and any attached files.

Reviewer #1: No

Reviewer #2: No

---

## [Author Response · Author response to Decision Letter 0]

6 Jan 2022

We very much appreciate the work of the Reviewers, which we feel has helped us to improve the paper substantially. We offer responses to their comments as follows.

1. The argument tries to be clear about how it is based on the data. We have tried to clarify the argument in places. We believe that our procedure is rigorous and sound.

2. We believe that our data and analysis are rigorous and have done what we can to check these.

3. Our data are fully available.

4. The manuscript has been substantially written, and thoroughly edited, by an experienced native speaker of English. We believe there are no major errors.

5. 

Reviewer #1: The comment is correct, that there is a relationship with another published paper, as noted [full citation: Mutambik I, Almuqrin A, Liu Y, Alhossayin M, Qintash FH. Gender Differentials on Information Sharing and Privacy Concerns on Social Networking Sites. J Glob Inf Manag. 2021 29(3):236–55. DOI 10.4018/JGIM.2021050110], which is closer than was anticipated at the time of submission. We have revised the manuscript (especially on pages 14-20, 27) to make this relationship clear and avoid duplication. The two papers are based on data from the same empirical study, but the focus, analysis and discussion are different. Specifically, the present paper includes in the analysis the construct of Attitude, which we see as important to the argument concerning culture, which does not arise in the other paper. The other paper is focussed on issues of gender: since these are not a central concern in the present paper, we have removed those sections which addressed gender specifically. We feel that this makes the present paper clearer in focus and in structure, as well as a little shorter. The discussion section in the paper is relatively long, but we feel this is necessary to explain how and why our perspective diverges from that which is commonly found in the literature, taking culture into account. We propose a revision of the title of the paper to help emphasise the focus of the paper, to read:

User concerns regarding information sharing on social networking sites: The user’s perspective in the context of national culture.

Reviewer #2: We recognise the importance of the points raised by this Reviewer, and the appropriateness of the additional references suggested, which we have been very happy to include and for which we are grateful. Our focus is on the concerns of academic users sharing information – perhaps they tend to take unduly for granted the accuracy of the information, and should think carefully about how it will be perceived, but these are large issues that would demand a very thorough discussion. In the paper, we have chosen to focus particularly on privacy because this is a particularly salient concern of the users we have studied. We argue there are cultural specificities in the nature of these concerns, but there is much work still to be done in this area and we do not have data of our own from other cultures for direct comparison. We have revised the manuscript to take the Reviewer’s suggestions into account and to try to clarify the focus and limitations of our discussion (mainly pages 3, 6-7, 45).

---

## [Editor Report · Decision Letter 1]

13 Jan 2022

User concerns regarding information sharing on social networking sites: The user’s perspective in the context of national culture

PONE-D-21-28070R1

,

We’re pleased to inform you that your manuscript has been judged scientifically suitable for publication and will be formally accepted for publication once it meets all outstanding technical requirements.

Kind regards,

Luigi Lavorgna

Academic Editor

PLOS ONE
---

## [Editor Report · Acceptance letter]

17 Jan 2022

PONE-D-21-28070R1 

User concerns regarding information sharing on social networking sites: The user’s perspective in the context of national culture 

Dear Dr. Lee:

I'm pleased to inform you that your manuscript has been deemed suitable for publication in PLOS ONE. Congratulations! Your manuscript is now with our production department. 

Kind regards, 

on behalf of

Dr. Luigi Lavorgna 

Academic Editor

PLOS ONE